# An Automated Real-Time PCR Assay versus Next-Generation Sequencing in the Detection of *BRAF* V600 Mutations in Melanoma Tissue Samples

**DOI:** 10.3390/diagnostics14151644

**Published:** 2024-07-30

**Authors:** Daniela Lenders, Irina Bonzheim, Matthias Hahn, Maximilian Gassenmaier, Valentin Aebischer, Andrea Forschner, Max Matthias Lenders, Lukas Flatz, Stephan Forchhammer

**Affiliations:** 1Department of Dermatology, University of Tuebingen, 72076 Tuebingen, Germanystephan.forchhammer@med.uni-tuebingen.de (S.F.); 2Department of Pathology, University of Tuebingen, 72076 Tuebingen, Germany

**Keywords:** melanoma, *BRAF* mutation, next-generation sequencing, Idylla^TM^

## Abstract

Background: Next-generation sequencing (NGS) is the most commonly used method for determining *BRAF* mutational status in patients with advanced melanoma. Automated PCR-based methods, such as the Idylla^TM^ system, are increasingly used for mutation diagnostics, but it is unclear what impact the choice of diagnostic method has on the management of melanoma. Objectives: To compare the concordance rate of *BRAF* V600 mutational analysis using Idylla^TM^ and NGS and to analyze the technical and clinical turnaround time. The clinical relevance is compared by analyzing the impact on the treatment decision. Methods: In this monocentric prospective cohort study, the *BRAF* mutation status of 51 patients was determined using both methods in parallel. Results: *BRAF* V600 mutation was detected in 23/51 cases (45%). Idylla^TM^ showed a 100% concordant result with a faster turnaround time (0.2 days) compared to NGS (12.2 days). In general, less tumor material was required for Idylla^TM^ than for NGS. Most patients received immunotherapy as a first-line therapy regardless of the *BRAF* V600 status. Conclusions: Idylla^TM^ testing proved to be a reliable and rapid alternative to NGS in the determination of *BRAF* V600 mutation. Although *BRAF.* status was available earlier, this had no influence on the treatment decision in most cases.

## 1. Introduction

The incidence of malignant melanoma is increasing worldwide. Historically, advanced melanoma had a poor prognosis due to the lack of effective treatment options as response rates to conventional chemotherapies are usually low. Better understanding of the molecular pathogenesis of melanoma has led to the discovery of new therapeutic options, such as targeted therapies [1] or immune checkpoint inhibitors [2]. As 50% of melanomas harbor activating mutations in the *BRAF* gene (90% affect codon V600E) with consequent activation of the MAPK-ERK-signaling pathway, BRAF inhibitors, such as vemurafenib, dabrafenib, encorafenib, in combination with MEK-inhibitors (cobimetinib, trametinib, binimetinib) play an important role in the treatment of advanced melanoma [3]. Targeted therapy with BRAF inhibitors results in a fast and direct therapy response with rare primary resistance, which is essential in the treatment of patients with rapid tumor progress, high tumor burden, or cerebral metastases [4]. The overall survival rate for therapy-naive BRAF-mutated advanced melanoma patients is 34% after 5 years of combined targeted therapy with BRAF/MEK-inhibitors, whereas patients receiving a first-line immune checkpoint blockade showed an overall 5-year survival rate of 60% [5].

Clinical testing for *BRAF* mutation may involve various testing systems, such as single-variant or multi-variant genotyping technologies. In diagnostic practice, various methods are used to determine the correct *BRAF* status, and choosing the appropriate diagnostic method is controversial. Single-variant analyses, such as the Idylla^TM^ assay specifically analyze mutations that can be used therapeutically and are, therefore, a cost- and time-effective method in the diagnostic setting. In contrast, NGS can simultaneously detect many other genomic mutations, although most of them do not have therapeutic consequences and are currently primarily identified for scientific purposes. This could potentially lead to new emerging therapeutic options in the time course.

In many instances, the adopted standard procedure is next-generation sequencing (NGS). NGS is characterized by a high sensitivity, high specificity, and the ability to analyze several genes simultaneously [6]. On the other hand, NGS requires a longer turnaround time compared with other, PCR-based methods [6,7] and is quite an expensive technique, requiring a substantial amount of equipment and a specialized laboratory [8]. The long turnaround time of NGS is a challenge in the clinical routine, as it forces clinicians to either choose to wait for the results or to start therapy with incomplete information. Nevertheless, NGS diagnostics is widely used, as it allows for the simultaneous analysis of multiple genes and is also performed in the routine diagnostic of other tumors. The Idylla^TM^ system (Biocartis, Belgium) is a fully automated real-time PCR (RT-PCR)-based test, designed to detect common *BRAF* mutations (*BRAF.* V600E, E2, D, K, R, and M) with high sensitivity and a fast turnaround time, providing rapid detection of *BRAF* mutation [8]. It is a CE-IVD-certified platform for the detection of *BRAF* mutations in formalin-fixed paraffin-embedded (FFPE) material.

In this study, we evaluated the Idylla^TM^ system for *BRAF* mutation testing in comparison to NGS in 51 melanoma patients, comparing concordance rates, turnaround times, and impact on treatment decisions.

## 2. Materials and Methods

### 2.1. Study Design

We conducted a monocentric prospective study at the university clinic of dermatology in Tuebingen, Germany, during the period from October 2020 to June 2021. We included adult patients (n = 51) with confirmed diagnosis of melanoma (any stage) with indication for *BRAF* mutation testing and unknown *BRAF.* mutation status. The indication for mutational analysis was defined by the dermato-oncologist treating the patient. All patients had to give their written informed consent for the participation in the study. Patient samples were obtained in the course of routine clinical excision of melanoma tissue, which was usually via surgery with local anesthesia. Excluded were patients with limited tumor material not allowing the parallel testing of two methods as well as underage patients and patients who did not give their informed consent. The decision as to whether the mutation analysis was performed from the primary tumor or metastasis was made by the responsible clinician. If too little tumor material was available in the requested block, an alternative tumor sample from the patient was used for comparative mutation analysis in consultation with the referring clinician. The following clinical parameters were collected from the electronic patient records (SAP i.s.h med system, version 2023) of the university hospital in Tuebingen: age, sex, location of the primary tumor, histological tumor subtype, presence of ulceration, Breslow tumor thickness/T stage, tumor stage (AJCC 2018), and therapy. Furthermore, we collected data in our laboratory regarding the tumor material used for analysis, the percentage of tumor cells, tumor surface area, DNA amount for NGS, number of sections used for the Idylla^TM^ analysis, detected mutations, allele frequencies, mean analytical time (time from preparation of FFPE section until the availability of the result), and mean clinical time (time from the data request of the oncologist until the result was delivered to the clinician).

### 2.2. Samples and Mutation Testing

The melanomas were diagnosed on H&E-stained section slides by board-certified dermatopathologists. If necessary, additional immunohistochemical stains were performed to confirm the diagnosis. The tumor borders were marked on the H&E-stained slide, and tumor size and tumor cell content were determined. The tumors were processed as 5 µm thick FFPE slides. A microdissection of the tumor was conducted and is especially recommended in the manufacturer’s protocol in samples with a low tumor cell content < 50%. Consecutive sections were tested for *BRAF* mutation using the Idylla^TM^
*BRAF.* mutation test on the CE-IVD certified Biocartis Idylla^TM^ System and the NGS method using the Ion Torrent NGS System Ion GeneStudio S5 as reference analysis. 

In one patient case with an urgent therapy indication due to a vital necessity, the Idylla^TM^ analysis was performed directly on fresh tumor tissue to obtain results more quickly. The result was then confirmed using FFPE material according to the study protocol.

### 2.3. Next-Generation Sequencing (NGS) Protocol

NGS analysis was carried out in cooperation with the institute of pathology at the university clinic in Tuebingen. Targeted multigene mutation screening was performed by next-generation sequencing (Ion GeneStudio S5, Thermo Fisher Scientific, Waltham, MA, USA) using an AmpliSeq Custom panel (hotspot regions in *BRAF.* exon 11, 15; *NRAS* exon 2, 3, 4; *KIT.* exon 8, 9, 11, 13, 14, 17, 18; *GNAQ* exon 5; *GNA11* exon 5; *CTNNB1* exon 3; *PDGFRA* exon 12, 13, 14, 18; *KRAS* exon 2, 3, 4, and *MAP2K1* exon 2, 3, 6, 7, 11). Amplicon library preparation and semiconductor sequencing was carried out according to the manufacturers’ manuals using the Ion AmpliSeq Library Kit v2.0, the Ion Library TaqMan Quantitation Kit on the Light-Cycler 480 (Roche, Basel, Switzerland), the Ion 540 Kit—Chef on the Ion Chef, and the Ion 540 Chip Kit (Thermo Fisher Scientific) [9]. For NGS analysis, a total DNA amount of 10 ng was used. Output files were generated with Torrent Suite 5.16. Variant calling of non-synonymous somatic variants compared to the human reference sequence hg19 was performed using Ion Reporter Software (Thermo Fisher Scientific, Version 5.18). Detection thresholds were set at an allele frequency of 5%. Variants identified by the Ion Reporter Software were visualized using the Integrative Genomics Viewer (IGV; Broad Institute, Cambridge, MA, USA; Version 2.11.9) to exclude panel-specific artefacts. The NCBI dbSNP database (including GnomAD, ExAC and TOPMED) was used to exclude SNPs.

### 2.4. Idylla^TM^ Protocol

The Idylla™ *BRAF* Mutation Test performed on the Biocartis Idylla™ System (Biocartis, Mechelen, Belgium) is a fully automatic, CE-IVD-certified diagnostic test with a rapid technical run time (approximately 150 min) for the qualitative detection of *BRAF.* V600E, E2, D, K, R and M mutations. The limit of detection in terms of allele frequency (AF) is 5%. During testing, disposable Idylla™ *BRAF* Mutation Test cartridges were loaded with FFPE human melanoma tissue sections without prior manual deparaffinization or FFPE preprocessing. Cartridges were inserted into the Idylla™ instrument, per the manufacturer’s instructions. The Idylla™ console and instruments were marked as being Conformité Européene. Inside the Idylla™ cartridge, deoxyribonucleic acid (DNA) was liberated from FFPE material using a combination of reagents, enzymes, heat, and high intensity-focused ultrasound. Within the cartridge, allele-specific multiplex PCR was performed for the amplification of specific mutated *BRAF* gene sequences. Conserved regions of the *BRAF* genes served as sample processing controls and as a measure of the amplifiable DNA in each sample, which is represented in each case by the quantitation cycle (Cq) value. The conserved regions were amplified in parallel with mutated *BRAF* gene sequences, where present. All required consumables were provided in the cartridge. The official in vitro diagnostic certified Idylla™ console report was used as the final Idylla™ result.

### 2.5. Statistics

Numerical variables were described by mean value and standard deviation (SD) or median and range using the program Microsoft Excel 2019. For statistical analysis, student’s *t*-tests were performed. A *p*-value ≤ 0.05 was considered as significant, while a *p*-value < 0.01 was considered statistically highly significant. Agreement between the Idylla™ Mutation Test and NGS as a comparator method was evaluated by estimating the overall, positive, and negative percentage diagnostic agreement (OPA, PPA, and NPA, respectively) together with the 95% two-sided Wilson score confidence interval (CI). 

### 2.6. Ethics Statement

Use of patient samples was approved by the local ethics committee (number 372/2020BO1) and was in accordance with the Declaration of Helsinki.

## 3. Results

### 3.1. Patient Characteristics

A total of 51 patients with advanced melanoma were included in the study. A total of 33 (65%) were female, while 18 (35%) were male. The mean age of the cohort was 67 years, and tumors were mainly localized on the extremities. Furthermore, 10 patients (20%) had occult metastatic melanoma, characterized solely by metastases without a primary tumor. In our cohort, there were no patients with mucosal or acral melanomas during this time period. Regarding the histological subtype, most patients had a nodular melanoma (18 patients, 35%) or superficial spreading melanoma (17 patients, 33%). Mean Breslow tumor thickness was 3.5 mm (SD: 3 mm) and ulceration was found in 33% of tumors. According to the AJCC melanoma staging system (8th edition 2018), 7 patients (14%) were classified stadium IV, 40 patients (78%) were stadium III, and 4 patients (8%) were stadium II. All epidemiological data are summarized in Table 1.

### 3.2. Technical Data

In 30 patients (59%), mutation analyses (NGS and Idylla^TM^) were performed of the primary tumor, which in 21 cases (41%) were of melanoma metastases (4/21 lymph node metastases and 17/21 cutaneous metastases). The mean percentage of tumor cells in the analyzed specimens was 81%, and mean tumor surface was 88 mm^2^. The mean DNA amount used for NGS was 56 ng/µL, and this was not assessed for Idylla^TM^ testing. An average of three slides were used for the Idylla^TM^ analysis compared to an average of six slides used for NGS. All technical data are summarized in Table 2.

### 3.3. Comparative Results of BRAF Testing Using NGS and the Idylla^TM^ Platform

In NGS analysis, we detected *BRAF* codon 600 mutations in 23 cases (45%) of which 21/23 (91%) harbored a p.V600E mutation and 2/23 (9%) harbored a p.V600K mutation. In the 23 cases of *BRAF* mutated melanoma samples, we found a mean allele frequency of 42 percent (range 6–84%).

The other cases (28/51, 55%) were negative for *BRAF* mutation. With NGS analysis, we were able to further elucidate the existing mutations in the *BRAF* wildtype cohort. The most common mutations in this cohort were *NRAS* mutations in 17/28 patients (33%). In 5/28 patients (10%), no mutation was detected. Other detected mutations included atypical *BRAF* mutation (1/28 patients, 2%), atypical *BRAF* and *NRAS* mutations (2/28 patients, 4%), *KRAS* mutation (1/28 patients, 2%), *MAP2K* mutation (1/28 patients, 2%), and an atypical *BRAF* and *MAP2K* mutation (1/28 patients, 2%). The Idylla^TM^ platform also identified all *BRAF* codon 600 mutations (PPA: 100%, CI: [1,1]) and all *BRAF*-negative (wild-type) cases (NPA: 100%, CI: [1,1]) showing a 100% concordance (OPA: 100%, CI: [1,1]) with NGS analysis as the reference method.

The mean analytical time from preparation of FFPE sections until availability of the result was 0.2 days (SD: 0.7 days) for Idylla^TM^ compared to 12.2 days (SD: 4.7 days) for NGS. This result was statistically highly significant (*p* = 5.43354 × 10^−24^). The mean clinical time from data request of the oncologist until the result was delivered with the Idylla^TM^
*BRAF* mutation platform was 5.5 days (SD: 6 days), in contrast to 17.5 days (SD: 7.3 days) using NGS. All results are summarized in Table 3. 

### 3.4. Treatment

Additionally, we analyzed the patients’ therapy in conjunction with their tumor stage and *BRAF* status (Figure 1). All four patients with tumor stage II were *BRAF* wildtype (100%) and all of the patients (100%) received no therapy (Figure 1a).

Out of the 40 patients with tumor stage III, 27 patients (67.5%) received immunotherapy, and 5 patients (12.5%) received targeted therapy (Figure 1b). Most of these patients received adjuvant therapy. Eight patients (20%) did not receive any therapy due to refusal or contraindications. In stage III patients, we found *BRAF* mutations in 18 (45%) patients. In the immunotherapy cohort in stage III, 19 patients (47.5%) were *BRAF* wild-type, and 8 patients (20%) had *BRAF* mutation.

Among the 32 stage III patients under therapy, treatment was initiated in 6 (19%) of the patients before the Idylla^TM^ result was available. Among the remaining 26 patients, treatment was initiated after a mean of 43 days (SD: 35 days) after the Idylla^TM^ result.

Of the seven stage IV melanoma patients, five patients (71%) received immunotherapy, one patient (14%) received targeted therapy, and one patient (14%) recieved no treatment (Figure 1c). Five of the patients (71%) had a detectable *BRAF* mutation. Out of the patients receiving immunotherapy in stage IV, three patients (43%) had *BRAF.* mutations, and the remaining two patients (29%) were *BRAF* wild-type. The patient receiving targeted therapy (14%) was *BRAF*-mutated, and the patient without therapy had a *BRAF* mutation as well.

In the case of patients in stage IV who received therapy, immunotherapy was initiated in one patient even though the results of the mutation analysis were pending. In the case of one patient initially presenting with an advanced melanoma with a high tumor burden, tumor fresh tissue was promptly collected on the same day, and an off-label Idylla^TM^ analysis was conducted out of the fresh tumor material outside of the study due to the vital threat. As the Idylla^TM^ result was available on the same day, this allowed the initiation of therapy directly with BRAF/MEK inhibitors as soon as confirmation of the presence of a *BRAF* mutation was established. The results were confirmed using FFPE material with Idylla^TM^ and NGS according to the study protocol. In this case, the prompt availability of the Idylla^TM^ analysis directly influenced the therapeutic approach. Among the remaining four stage IV patients, therapy was initiated after a mean of 22.75 days (SD: 28.5 days) after the Idylla^TM^ result.

## 4. Discussion

The determination of the *BRAF* codon 600 mutation status is essential in the treatment of patients with advanced melanoma. In diagnostic practice, various methods are used to determine the correct *BRAF* mutation status, and choosing the appropriate diagnostic method remains controversial [10].

In this single-center prospective study, we compared the performance and clinical relevance of a rapid, fully automated real-time PCR assay (Idylla^TM^ system) with NGS as the standard procedure at our clinic in a cohort of 51 melanoma patients with an indication for *BRAF* mutation testing.

Regarding the epidemiological data, a notable aspect in our cohort is the relatively high number of patients with occult metastatic melanoma (20%). Contrary to data indicating an overall incidence of 2% [11], this could be attributed to the fact that patients with an occult melanoma inherently present at a higher tumor stage, thereby warranting an indication for *BRAF* mutation analysis. Furthermore, our group showed an above-average presence of nodular and ulcerated melanomas, leading to a higher T-stage and, consequently, an advanced tumor stage, necessitating *BRAF* analysis. In our group, no lentigo maligna melanomas were found, possibly indicating that these tend to have a longer growth period, resulting in a frequently lower tumor thickness and stage.

Looking at the technical data, it is noticeable that fewer sections were required for Idylla^TM^ analysis compared to NGS. This shows that Idylla^TM^ can be performed even with limited tumor material, as has been demonstrated in other studies [12]. This could be attributed to the fact that Idylla^TM^ analysis does not require an additional DNA extraction step with potential material loss in contrast to NGS.

In our study, we found a 100% concordance of the results of Idylla^TM^ and NGS. This high sensitivity and reliability have also been demonstrated in other studies involving melanoma and various other tumor types [8,12,13]. A significant difference between the two methods is the markedly reduced turnaround time of Idylla^TM^ compared to NGS. In our study, the mean analytical time was 0.2 days with Idylla^TM^ compared to 12.2 days with NGS. Additionally, we analyzed the clinical turnaround time for both methods, referring to the duration from the oncologist’s request to the communication of findings to the clinician, which was 5.5 days for Idylla^TM^ and 17.5 days for NGS. This difference in analytical and clinical time is due to additional steps in the clinical process, such as requesting external tumor materials or the preparation of FFPE material. The reduced turnaround time for Idylla^TM^ in comparison to NGS is consistent with data from other studies. Previously, FACILITATE, a real-world, prospective, multicenter, European study, evaluated the performance and analytical turnaround time of the Idylla^TM^ *EGFR* mutation test compared with local reference methods in non-small-cell-lung-cancer patients, showing an overall percentage agreement of 97.7% and an analytical turnaround time within 1 week versus about 22 days using reference methods [13]. The faster availability of mutation results can offer advantages in the patient’s clinical care. Surprisingly, our study showed that the faster availability of mutation analysis is only necessary in very few cases. Thus, the faster availability of *BRAF* codon 600 mutation analysis had a direct influence on the treatment decision in only 1 of the 51 cases examined. In our cohort, one patient was initially diagnosed with advanced melanoma and progressive brain metastases. In such situations, clinicians often face the decision of waiting for the correct result, causing a delay in initiating therapy, or starting treatment despite incomplete results. As BRAF/MEK inhibitor therapy tends to produce faster and higher tumor shrinkage rates in the beginning than immunotherapy, in patients with high tumor burden therapy is often initiated with targeted therapy if *BRAF* mutation is detectable [14,15]. In our patient, we extracted tumor tissue and conducted the Idylla^TM^ analysis directly on fresh tissue in addition to our study on FFPE material. This approach allowed us to obtain the Idylla^TM^ result on the same day without waiting for the tissue fixation and embedding steps, and we were able to start targeted therapy right on the very same day after receiving the result that the patient did indeed have a *BRAF* mutation. The result from the fresh tissue was confirmed by using Idylla^TM^ and NGS on FFPE material. The approach of conducting the Idylla^TM^ analysis directly from fresh tumor tissue allows for mutation analysis with minimal time delay. We recommend evaluating this approach with further studies of Idylla^TM^ on fresh tumor tissue, as until now it is only approved for FFPE material [12].

A drawback of the Idylla^TM^ method is that only selected *BRAF* and therapeutically relevant *BRAF.* mutations can be detected, without the ability to identify further mutations. Up to now, only those patients with a typical *BRAF.* mutation (*BRAFV*600) are primarily treated with BRAF/MEK-inhibition. Furthermore, there are atypical *BRAF.* mutations which cannot be detected using Idylla^TM^. Menzer et al. observed that targeted therapy with BRAF/MEK inhibitors could also elicit a response in some cases of atypical *BRAF* mutations in a retrospective study [16]. However, this response was lower compared to melanomas with *BRAF*V600 mutations. For comprehensive assessments of all mutations present or when quantitative information of allele frequency is required, it is advisable to employ other more specialized molecular techniques, such as NGS [17]. One example is for mucosal or acral melanomas, which occasionally exhibit a *KIT* mutation that provides a therapeutic target for imatinib. For these melanoma types, additional methodologies, such as NGS, should be employed. However, in our study, these melanoma types did not occur during this period.

Additionally, we investigated the therapy of patients, depending on the tumor stage and the *BRAF* result. It is noticeable that *BRAF* mutation analysis was conducted in four patients at tumor stage II. However, at that time, these patients did not receive any therapy, as adjuvant therapy was not yet approved for stage II. This changed with the approval of adjuvant therapy in melanoma patients in completely resected melanoma stage IIB/IIC in the year 2022, based on the data from the KEYNOTE-716 study [18]. However, it is important to note that the approval is specifically for adjuvant immunotherapy and not for targeted therapy. Nonetheless, we consider an early mutation analysis to be meaningful, especially since stage IIB and IIC involve high-risk melanomas with a high risk for tumor progression over the course of the disease [19]. When considering patients in stage III in our study, more patients, despite having a *BRAF* mutation, received (mostly adjuvant) immunotherapy instead of targeted therapy. It was notable that in 6 out of 32 patients in stage III, adjuvant therapy was started without knowing the mutation analysis result. In this cases therapy was initiated directly with immunotherapy, since both therapies (immuno- and targeted therapy) are considered as equivalent adjuvant treatment options [20,21,22]. On the other hand, for the remaining patients, it was notable that there was a significant time delay until the initiation of therapy, despite the prompt availability of the Idylla^TM^ result. Therefore, it can be concluded that the Idylla^TM^ analysis had no impact on the clinical treatment decision for stage III patients in our study. Similar observations were made in stage IV as well. In this stage, despite having a *BRAF* mutation, most patients initially received immunotherapy instead of targeted therapy. In this scenario, the results of the mutation analysis were also not waited for with one patient, and therapy was initiated directly with immunotherapy. In the case of the other patients, a time delay until the initiation of therapy was also observed, suggesting that the quick availability of the mutation result would not have been necessary in most cases within our cohort, as in most patients it had no direct impact on the therapeutic approach. Especially in cases with a relatively low tumor burden in stage IV, primary immunotherapy (often as combined immunotherapy with ipilimumab and nivolumab) is frequently favored in our clinic to achieve a long-term therapy response. This approach was recently examined in the DREAMseq Trial, revealing a superiority on the overall survival (OS) of primary immunotherapy over primary targeted therapy [15] in *BRAF*-mutated progressive melanoma patients. In only one patient in our study with high tumor burden and imminent vital threat did the immediate availability of the mutation result with Idylla^TM^ have a significant impact, namely the direct start with targeted therapy, as described above.

Our study has some limitations. Firstly, it was a monocentric study with a relatively small cohort of 51 patients. This limits the significance of the comparative agreement. In addition, the monocentric setting reflects the approach of a single clinic in particular, which limits the generalizability of our results. Most of the patients were in stage III (lymph node metastases). Only a few patients were in stage IV with an urgent need for treatment. It is possible that in a cohort with more patients with urgent treatment initiation, the effects on the treatment decision would have been more evident.

In summary, Idylla^TM^ mutation analysis provided a comparable but only qualitative result of the *BRAF* mutation status compared to NGS examination. Clinically, this had no direct consequence on the patients’ treatment in most cases. An advantage of Idylla^TM^ is the quicker availability of the mutation result, especially in patients facing a vital threat, where a delayed initiation of therapy can lead to a poorer prognosis. Therefore, we recommend the use of Idylla^TM^ in such patient scenarios, and we would like to refer to our approach of analyzing fresh material, which, however, should be further investigated in additional studies.

## Figures and Tables

**Figure 1 diagnostics-14-01644-f001:**
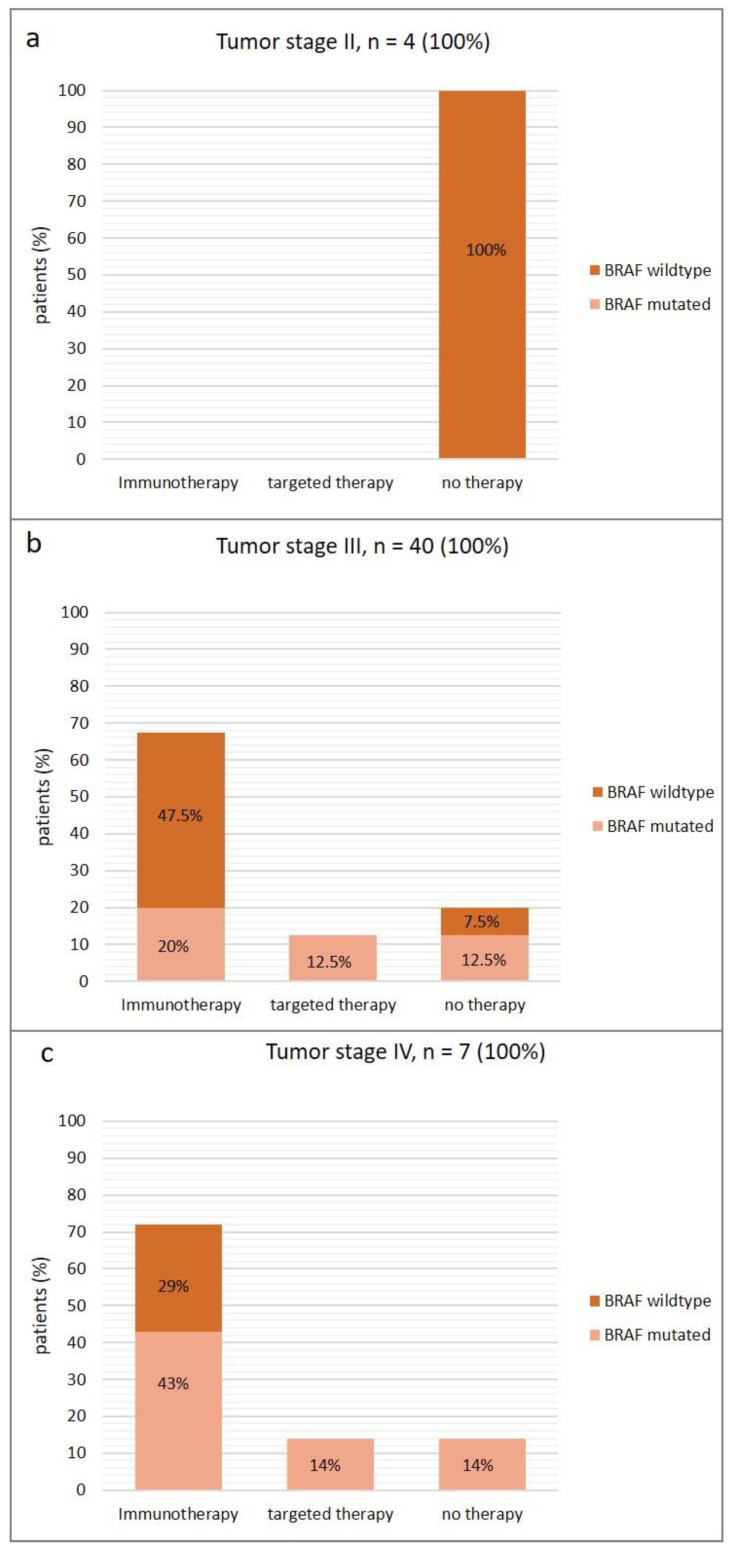
Patients´ therapy in conjunction with their tumor stage ((**a**): stage II, (**b**): stage III, (**c**): stage IV) and the *BRAF* mutation status.

**Table 1 diagnostics-14-01644-t001:** Patient characteristics.

Characteristics	Value
**Total, n**	51
**Age, years, mean, (SD; range)**	67 (14; 31–92)
**Sex, n (%)**	
Male	18 (35)
Female	33 (65)
**Localization of primary tumor, n (%)**	
Trunk	9 (18)
Head	2 (4)
Arms	13 (25)
Legs	17 (33)
Occult	10 (20)
**Tumor subtype, n (%)**	
Nodular melanoma	18 (35)
Superficial spreading melanoma	17 (33)
Acral lentiginous melanoma	3 (6)
Occult melanoma	10 (20)
No subtype specified	3 (6)
**Ulceration, n (%)**	
Yes	17 (33)
No	23 (45)
Not specified	11 (22)
**Breslow tumor thickness, mm, mean (SD; range)**	3.5 (3; 0.9–17)
**Breslow tumor thickness/T-stage (TNM classification, AJCC 2018), number of patients (%)**	
≤1.0 mm/T1	2 (4)
>1.0–2.0 mm/T2	14 (27)
>2.0–4.0 mm/T3	11 (22)
>4.0 mm/T4	13 (25)
Occult/no T-stage specified	11 (22)
**Melanoma stage (8th edition, AJCC 2018)**	
II, n (%)	4 (8)
III, n (%)	40 (78)
IV, n (%)	7 (14)

**Table 2 diagnostics-14-01644-t002:** Technical data.

Characteristics	Number
**Tumor tissue**	
Primary tumor n (%)	30 (59)
Metastases, n (%), thereof	21 (41)
- Lymph node metastases, n (%)	4 (19)
- Cutaneous metastases, n (%)	17 (81)
**Percentage of tumor cells,** **mean (SD; range)**	81 (15; 15–95)
**Tumor surface in mm^2^** **, mean (SD; range)**	88 (77; 10–400)
**DNA amount for NGS** **, ng/µL, mean (SD; range)**	56 (30; 5–120)
**Number of sections, mean (SD; range)**	
NGS	6 (5; 1–20)
Idylla^TM^	3 (3; 1–14)

**Table 3 diagnostics-14-01644-t003:** Comparative results of *BRAF* testing using NGS and Idylla^TM^.

Characteristics	NGS	Idylla^TM^
***BRAF* mutated, n (%)**	23 (45)	23 (45)
- Allele frequency in %, mean (SD; range)	42 (20; 6–84)	
*BRAF* p.V600E	21 (91)	
*BRAF* p.V600K	2 (9)	
***BRAF* wildtype, n (%)**	28 (55)	28 (55)
*NRAS* mutation	17 (33)	
Atypical *BRAF* mutation	1 (2)	
- *BRAF* p.G496R	1	
Atypical *BRAF* and *NRAS* mutation	2 (4)	
- *BRAF* p.L584F +*NRAS* p.Q61R	1	
- *BRAF* p.K601N + *NRAS* p.Q61R	1	
*KRAS* mutation	1 (2)	
*MAP2K* mutation	1 (2)	
Atypical *BRAF* and *MAP2K* mutation	1 (2)	
- *BRAF* p.K601E + *BRAF* p.L584F + *MAP2K1* p.P124S	1	
No mutation detected	5 (10)	
**Mean analytical time, days (SD)**	12.2 (4.7)	0.2 (0.7)
**Mean clinical time, days (SD)**	17.5 (7.3)	5.5 (6)

## Data Availability

The data sets analyzed during the current study are available from the corresponding author on reasonable request.

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
