# Peer review of "An Automated Real-Time PCR Assay versus Next-Generation Sequencing in the Detection of BRAF V600 Mutations in Melanoma Tissue Samples"

_diagnostics, 2024, doi:10.3390/diagnostics14151644_

Round 1

Reviewer 1 Report

Comments and Suggestions for Authors

Dear Editor,

I read the article "An automated real-time PCR assay versus NGS in the detection of BRAF V600 mutations in melanoma tissue samples" by Daniela Lenders, Irina Bonzheim, Matthias Hahn, Maximilian Gassenmaier, Valentin Aebischer, Andrea Forschner, Max Matthias Lenders, Lukas Flatz, and Stephan Forchhammer from the University of Tuebingen, Germany, with great interest.

The primary objective of the study was to compare the concordance rate of BRAF V600 mutational analysis using the IdyllaTM system, an automated PCR-based method, with Next Generation Sequencing (NGS). The study aimed to analyze the technical and clinical turnaround times and to evaluate the clinical relevance by examining the impact on treatment decisions.

The main result of the study indicated that the BRAF V600 mutation was detected in 23 out of 51 cases (45%). The IdyllaTM system showed a 100% concordance rate with the NGS results. Furthermore, the IdyllaTM system demonstrated a significantly faster turnaround time of 0.2 days compared to 12.2 days for NGS. The study concluded that while the IdyllaTM system provided reliable and rapid detection of BRAF V600 mutations, the earlier availability of BRAF status did not influence the treatment decision in most cases.

The article is very well written, and I highly recommend its publication. Below, I provide a checklist of the necessary structures for publication, which I evaluated in the article, with some minor adjustments that I consider necessary:

  • Abstract: Clearly summarizes the study as a whole, being representative of what was done without exaggerated conclusions or conclusions not supported by the presented results.
  • Introduction: Sets the problems, describes the scenario, and highlights the importance of the study. It establishes the differences between the compared methods and states each method's advantages and disadvantages. The study objective is clearly defined at the end of the introduction section.
  • Materials and Methods: Describes the study design, study period, and from where the patients were enrolled. It details the inclusion and exclusion criteria, ethics approval, how melanomas were diagnosed, how the slides were prepared, NGS targets beyond BRAF, and the tested methods (BRAF V600 mutational analysis using IdyllaTM). Detailed procedures are provided in the supplement.
    • Suggestion 1: As this is a somatic analysis, it would be valuable to declare the limits of detection of each compared technology in the Methods section.
    • Suggestion 2: Another suggestion is to ensure that all variables described and analyzed in the results have a description in the methods, specifically from where the data were obtained and how they were analyzed. This includes information about metastasis, tumor classification, treatment/therapy, other mutations (beyond BRAF), TAT of the assays, and the case described where the tested methods were applied to the fresh tissue. Readers must be prepared that these variables will be addressed in the results section, as some of them appear for the first time only in the results section.
  • Results: Presents the findings in a clear and organized manner. It includes patient demographic and epidemiological characteristics, melanoma characteristics, and the percentage of tumor cells, tumor surface, among others. Tables 1, 2, and 3 are well prepared and connect with the text without unnecessary repetition. Treatment details are well described, as are the BRAF and other gene mutations. Additionally, the main result is the phrase at line 142 which presents the global results of the comparison, showing 100% concordance between both tests.
    • Suggestion 3: As already suggested in the Materials and Methods commentaries, it is unclear from where the information about metastasis, tumor classification, treatment/therapy, other mutations (beyond BRAF), TAT of the assays, and the case described where the tested methods were applied to the fresh tissue comes, and it should be described in the Methods section. These variables are cited for the first time only in the results, so please ensure that all variables described have support in the Methods section.
    • Suggestion 4: A table with BRAF mutations VAF from NGS and Cq values for qPCR would be valuable information for the reader.
    • Suggestion 5: The total agreement between both tests is presented in line 142, but it can be broken down into total, positive, and negative agreement, which are common values described in methods head-to-head comparison. All values will be 100%. The authors could include the 95% CI of the percentage of agreement. The 95% CI will be wide and dependent on the sample size, but it is valuable information for the reader and adheres to the highest scientific description standards.
  • Discussion: Is well-structured and all presented results are discussed without exaggeration on conclusions or conclusions without results support.

Author Response

We thank Reviewer 1 for their suggestions and will address their comments below:

Suggestion 1: As this is a somatic analysis, it would be valuable to declare the limits of detection of each compared technology in the Methods section.

Response 1: We agree with the reviewer that the limitations of both methods should be mentioned in the methods section. Therefore, we have added a detailed description of the NGS and Idylla methods to this section and specified the exact genetic mutations detectable with each method:

“2.3.       Next generation sequencing (NGS) protocol

NGS analysis was carried out in cooperation with the institute of pathology at the university clinic Tuebingen. Targeted multigene mutation screening was performed by Next Generation Sequencing (Ion GeneStudio S5, Thermo Fisher Scientific, Waltham, MA, USA) using an AmpliSeq Custom panel (hotspot regions in BRAF exon 11, 15; NRAS exon 2, 3, 4; KIT exon 8, 9, 11, 13, 14, 17, 18; GNAQ exon 5; GNA11 exon 5; CTNNB1 exon 3; PDGFRA exon 12, 13, 14, 18; KRAS exon 2, 3, 4 and MAP2K1 exon 2, 3, 6, 7, 11). Amplicon library preparation and semiconductor sequencing was done according to the manufacturers’ manuals using the Ion AmpliSeq Library Kit v2.0, the Ion Library TaqMan Quantitation Kit on the Light-Cycler 480 (Roche, Basel, Switzerland), the Ion 540 Kit – Chef on the Ion Chef and the Ion 540 Chip Kit (Thermo Fisher Scientific) [9]. For NGS analysis, a total DNA amount of 10 ng was used. Output files were generated with Torrent Suite 5.16. Variant calling of non-synonymous somatic variants compared to the human reference sequence hg19 was performed using Ion Reporter Software (Thermo Fisher Scientific, Version 5.18). Detection thresholds were set at an allele frequency of 5%. Variants called by the Ion Reporter Soft-ware were visualized using the Integrative Genomics Viewer (IGV; Broad Institute, Cambridge, MA; Version 2.11.9) to exclude panel-specific artefacts. The NCBI dbSNP database (including GnomAD, ExAC and TOPMED) was used to exclude SNPs.

2.4.         IdyllaTM protocol

The Idylla™ BRAF Mutation Test performed on the Biocartis Idylla™ System (Biocartis, Belgium) is a fully automatic, CE-IVD certified diagnostic test with a rapid technical run time (approximately 150 min) for the qualitative detection of BRAF V600E, E2, D, K, R and M mutation. The limit of detection in terms of allele frequency (AF) is 5%. During testing, disposable Idylla™ BRAF Mutation Test cartridges were loaded with FFPE human melanoma tissue sections without prior manual deparaffinization or FFPE preprocessing. Cartridges were inserted into the Idylla™ instrument, per the manufacturer’s instructions. The Idylla™ console and instruments were Conformité Européene marked. Inside the Idylla™ cartridge, deoxyribonucleic acid (DNA) was liberated from FFPE material using a combination of reagents, enzymes, heat and high intensity-focused ultrasound. Within the cartridge, allele-specific multiplex PCR was performed for the amplification of specific mutated BRAF gene sequences. Conserved regions of the BRAF genes served as sample processing controls and as a measure of the amplifiable DNA in each sample, which is represented in each case by the quantitation cycle (Cq) value. The conserved regions were amplified in parallel with mutated BRAF gene sequences, where present. All required consumables were provided in the cartridge. The official in vitro diagnostic certified Idylla™ console report was used as the final Idylla™ result.”

Suggestion 2: Another suggestion is to ensure that all variables described and analyzed in the results have a description in the methods, specifically from where the data were obtained and how they were analyzed. This includes information about metastasis, tumor classification, treatment/therapy, other mutations (beyond BRAF), TAT of the assays, and the case described where the tested methods were applied to the fresh tissue. Readers must be prepared that these variables will be addressed in the results section, as some of them appear for the first time only in the results section.

Response 2: We thank the reviewer for this suggestion and have now ensured that all analyzed variables are described in the methods section:

The following clinical parameters were collected from the electronic patient records (SAP i.s.h med system, version 2023) of the university hospital Tuebingen: age, sex, location of the primary tumor, histological tumor subtype, presence of ulceration, Breslow tumor thickness/T stage, tumor stage (AJCC 2018) and therapy. Furthermore, we collected data in our laboratory regarding the tumor material used for analysis, percentage of tumor cells, tumor surface area, DNA amount for NGS, number of sections used for the IdyllaTM analysis, detected mutations, allele frequencies, mean analytical time (time from preparation of FFPE section until availability of the result) and mean clinical time (time from data request of oncologist until the result was delivered to the clinician).“

Additionally, we mentioned the patient case in which the analysis was conducted on fresh tissue:

“In one patient case with an urgent therapy indication due to a vital necessity, the IdyllaTM analysis was performed directly on fresh tumor tissue to obtain results more quickly. The result was then confirmed using FFPE material according to the study protocol.”

Results: Presents the findings in a clear and organized manner. It includes patient demographic and epidemiological characteristics, melanoma characteristics, and the percentage of tumor cells, tumor surface, among others. Tables 1, 2, and 3 are well prepared and connect with the text without unnecessary repetition. Treatment details are well described, as are the BRAF and other gene mutations. Additionally, the main result is the phrase at line 142 which presents the global results of the comparison, showing 100% concordance between both tests.

Suggestion 3: As already suggested in the Materials and Methods commentaries, it is unclear from where the information about metastasis, tumor classification, treatment/therapy, other mutations (beyond BRAF), TAT of the assays, and the case described where the tested methods were applied to the fresh tissue comes, and it should be described in the Methods section. These variables are cited for the first time only in the results, so please ensure that all variables described have support in the Methods section.

Response 3: We made sure, that all variables as well as the case with fresh tumor material are described in the methods section.

Suggestion 4: A table with BRAF mutations VAF from NGS and Cq values for qPCR would be valuable information for the reader.

Response 4: We added the allele frequencies from NGS in table 3 and in the text: “In the 23 cases of BRAF mutated melanoma samples, we found a mean allele frequency of 42 percent (range 6 – 84%).

 The Cq values were indeed recorded during the measurement, but they were not captured by the study protocol. Since the Idylla platform was a loaned device from the company Biocartis, it was not connected to our network, and the values were not uploaded to the Idylla-Cloud. Unfortunately, the loaned device has now been returned, and the data can no longer be retrieved. Unfortunately, we are therefore unable to provide the Cq values of the analysis.

Suggestion 5: The total agreement between both tests is presented in line 142, but it can be broken down into total, positive, and negative agreement, which are common values described in methods head-to-head comparison. All values will be 100%. The authors could include the 95% CI of the percentage of agreement. The 95% CI will be wide and dependent on the sample size, but it is valuable information for the reader and adheres to the highest scientific description standards.

Response 5: We have reported the values for total, positive, and negative agreement, as well as the 95% confidence interval, in the results section:

“The IdyllaTM platform also identified all BRAF codon 600 mutations (PPA: 100%, CI: [1,1]) and all BRAF negative (wild type) cases (NPA: 100%, CI: [1,1]) showing a 100 % concordance (OPA: 100%, CI: [1,1]) with NGS analysis as the reference method.”

Furthermore, we added the following to the methods section:

“Agreement between the Idylla™ Mutation Test and NGS as comparator method was evaluated by estimating the overall, positive and negative percentage diagnostic agreement (OPA, PPA and NPA, respectively) together with 95% two-sided Wilson score confidence interval (CI).”

Reviewer 2 Report

Comments and Suggestions for Authors

REVISION AND COMMENTS:

1. This work is of particular interest, since it describes a comparison between the gold standard method for detect common BRAF mutations (BRAF V600E, E2, D, K, R, M) in patients with advanced melanoma, notably, NGS, and a novel Automated real time PCR-based method named Idylla platform already employed in the clinical setting with FFPE samples. A total of 51 patients were screened for the presence of BRAF mutation status. Main results of the study indicate that IdyllaTM was 100% concordant with NGS, while (i) presenting a faster turnaround time, (ii) requiring less tumor material for the analysis. 

2. I believe that this study is well conducted, while this manuscript is well written and clear other than easy to follow. 

3. Please include in the introduction the 5-years survival rate of BRAF-mutated melanoma patients. PMID: 32124332

4. the study aim at 62-66 is quite redundant in both sentences, I suggest rephrasing this section in order to improve its readability

5. References should be incdelude in the methods. For instance, for NGS analysis this reference should be included PMID: 32302601, while for statistical analysis authors should include this work PMID: 32998835

6. "Although detailed descriptions are provided in the supplementary material, a specific section in the methods should be included for both NGS and Idylla. Considering that Idylla platform is the main focus of this work, it is particularly important to include a brief description of this technique in the methods section.

7. Why tumor tissue characteristics an other clinical information depicted in section 3.2 and table 2 are considered as “technical data”? I consider those clinical data of patients

8. Please include some limitations of the study. One limitation could be the relatively small number of patients included for this comparative analysis

Author Response

We would like to thank Reviewer 2 and will address their suggestions in detail below.

Suggestion 1: Please include in the introduction the 5-years survival rate of BRAF-mutated melanoma patients. PMID: 32124332

Response 1: We included the 5-year-survival rate of BRAF-mutated melanoma patients in the introduction section: “The overall survival rate for therapy naive BRAF mutated advanced melanoma patients is 34% after 5 years of combined targeted therapy with BRAF/MEK-inhibitors, whereas patients receiving first-line immune checkpoint blockade showed an overall 5-year survival rate of 60% [5].”

Suggestion 2: the study aim at 62-66 is quite redundant in both sentences, I suggest rephrasing this section in order to improve its readability

Response 2: We agree with the reviewer and have shortened and revised this passage as follows: “In this study, we evaluated the IdyllaTM system for BRAF mutation testing in comparison to NGS in 51 melanoma patients, comparing concordance rates, turnaround times, and impact on treatment decisions.”

Suggestion 3: References should be included in the methods. For instance, for NGS analysis this reference should be included PMID: 32302601, while for statistical analysis authors should include this work PMID: 32998835

Response 3: We thank the reviewer for suggesting the paper “PMID: 32302601” and have added this work to the materials and methods section of the NGS analysis:

“Amplicon library preparation and semiconductor sequencing was done according to the manufacturers’ manuals using the Ion AmpliSeq Library Kit v2.0, the Ion Library TaqMan Quantitation Kit on the Light-Cycler 480 (Roche, Basel, Switzerland), the Ion 540 Kit – Chef on the Ion Chef and the Ion 540 Chip Kit (Thermo Fisher Scientific) [9].”

Under the specified PMID: 32998835, we found the following paper: Corazza M, et al. Tissue cytokine/chemokine profile in vulvar lichen sclerosus: An observational study on keratinocyte and fibroblast cultures. J Dermatol Sci. 2020 Dec;100(3):223-226. Unfortunately, we cannot identify any overlaps with our project, especially since only descriptive statistics without citable literature were used in our paper. We have therefore decided not to add this reference to the methods section of our manuscript.

Suggestion 4: Although detailed descriptions are provided in the supplementary material, a specific section in the methods should be included for both NGS and Idylla. Considering that Idylla platform is the main focus of this work, it is particularly important to include a brief description of this technique in the methods section.

Response 4: We agree with the reviewer that the detailed presentation of the two methods should be given more space in the materials and methods section. For this reason, we have moved the exact protocols from the supplemental material section for both NGS and Idylla to the methods section.

“2.3.       Next generation sequencing (NGS) protocol

NGS analysis was carried out in cooperation with the institute of pathology at the university clinic Tuebingen.Targeted multigen mutation screening was performed by Next Generation Sequencing (Ion GeneStudio S5, Thermo Fisher Scientific, Waltham, MA, USA) using an AmpliSeq Custom panel (hotspot regions in BRAF exon 11, 15; NRAS exon 2, 3, 4; KIT exon 8, 9, 11, 13, 14, 17, 18; GNAQ exon 5; GNA11 exon 5; CTNNB1 exon 3; PDGFRA exon 12, 13, 14, 18; KRAS exon 2, 3, 4 and MAP2K1 exon 2, 3, 6, 7, 11). Amplicon library preparation and semiconductor sequencing was done according to the manufacturers’ manuals using the Ion AmpliSeq Library Kit v2.0, the Ion Library TaqMan Quantitation Kit on the Light-Cycler 480 (Roche, Basel, Switzerland), the Ion 540 Kit – Chef on the Ion Chef and the Ion 540 Chip Kit (Thermo Fisher Scientific) [9]. For NGS analysis, a total DNA amount of 10 ng was used. Output files were generated with Torrent Suite 5.16. Variant calling of non-synonymous somatic variants compared to the human reference sequence hg19 was performed using Ion Reporter Software (Thermo Fisher Scientific, Version 5.18). Detection thresholds were set at an allele frequency of 5%. Variants called by the Ion Reporter Soft-ware were visualized using the Integrative Genomics Viewer (IGV; Broad Institute, Cambridge, MA; Version 2.11.9) to exclude panel-specific artefacts. The NCBI dbSNP database (including GnomAD, ExAC and TOPMED) was used to exclude SNPs.

2.4.         IdyllaTM protocol

The Idylla™ BRAF Mutation Test performed on the Biocartis Idylla™ System (Biocartis, Belgium) is a fully automatic, CE-IVD certified diagnostic test with a rapid technical run time (approximately 150 min) for the qualitative detection of BRAF V600E, E2, D, K, R and M mutation. The limit of detection in terms of allele frequency (AF) is 5%. During testing, disposable Idylla™ BRAF Mutation Test cartridges were loaded with FFPE human mela-noma tissue sections without prior manual deparaffinization or FFPE preprocessing. Car-tridges were inserted into the Idylla™ instrument, per the manufacturer’s instructions. The Idylla™ console and instruments were Conformité Européene marked. Inside the Idylla™ cartridge, deoxyribonucleic acid (DNA) was liberated from FFPE material using a combination of reagents, enzymes, heat and high intensity-focused ultrasound. Within the cartridge, allele-specific multiplex PCR was performed for the amplification of specific mutated BRAF gene sequences. Conserved regions of the BRAF genes served as sample processing controls and as a measure of the amplifiable DNA in each sample, which is represented in each case by the quantitation cycle (Cq) value. The conserved regions were amplified in parallel with mutated BRAF gene sequences, where present. All required consumables were provided in the cartridge. The official in vitro diagnostic certified Idylla™ console report was used as the final Idylla™ result.”

Suggestion 5: Why tumor tissue characteristics and other clinical information depicted in section 3.2 and table 2 are considered as “technical data”? I consider those clinical data of patients

Response 5: The points presented in table 2, such as the tumor characteristics, were considered technical data, as the genomic analysis was performed using this material (for example, from the primary tumor or from the metastases). This information regarding the material from which the analysis was performed does not necessarily allow conclusions about the tumor stage of the patients, as in some cases, the analysis was conducted using the primary tumor in patients with distant metastases, if no material was available from the metastases. In contrast, the relevant clinical data such as the tumor stage of the patients is listed in table 1 (patient characteristics).

To clarify this issue, we made the following explanation to the manuscript: “In 30 patients (59 %), mutation analyses (NGS and IdyllaTM) were performed of the primary tumor, in 21 cases (41 %) out of melanoma metastases (thereof 4/21 lymph node metastases, 17/21 cutaneous metastases).”

Furthermore, we added the following passage to the material and methods section: “The decision as to whether the mutation analysis was performed from the primary tumor or metastasis was made by the responsible clinician. If too little tumor material was available in the requested block, an alternative tumor sample from the patient was used for comparative mutation analysis in consultation with the referring clinician.”

Suggestion 6: Please include some limitations of the study. One limitation could be the relatively small number of patients included for this comparative analysis.

Response 6: We agree with the reviewer that the study's limitations should be addressed and have incorporated the following changes into the discussion:

“Our study has some limitations. Firstly, it was a monocentric study with a relatively small cohort of 51 patients. This limits the significance of the comparative agreement. In addition, the monocentric setting reflects the approach of a single clinic in particular, which limits the generalizability of our results. Most of the patients were in stage III (lymph node metastases). Only a few patients were in stage IV with an urgent need for treatment. It is possible that in a cohort with more patients with urgent treatment initiation, the effects on the treatment decision would have been more evident.”